# Somatic Hypomethylation of Pericentromeric SST1 Repeats and Tetraploidization in Human Colorectal Cancer Cells

**DOI:** 10.3390/cancers13215353

**Published:** 2021-10-26

**Authors:** Beatriz González, Maria Navarro-Jiménez, María José Alonso-De Gennaro, Sanne Marcia Jansen, Isabel Granada, Manuel Perucho, Sergio Alonso

**Affiliations:** 1Program of Predictive and Personalized Medicine of Cancer, Germans Trias i Pujol Research Institute (PMPPC-IGTP), 08916 Badalona, Spain; bgonzalez@igtp.cat (B.G.); mnavarroj@igtp.cat (M.N.-J.); majosealonso97@gmail.com (M.J.A.-D.G.); sannemjansen@gmail.com (S.M.J.); 2Cytogenetics Platform, Hematology Laboratory Service, Institut Català d’Oncologia, Hospital Germans Trias i Pujol, Josep Carreras Leukaemia Research Institute, 08916 Badalona, Spain; igranada@iconcologia.net; 3Sanford Burnham Prebys Medical Discovery Institute (SBP), La Jolla, CA 92037, USA; mperucho@sbpdiscovery.org

**Keywords:** genomic DNA hypomethylation, colorectal cancer, tetraploidization, chromosome instability, repetitive elements, polyploidy

## Abstract

**Simple Summary:**

Cancer cells frequently exhibit an abnormal number of chromosomes, termed aneuploidy, often preceded by an aberrant genome duplication resulting in cells with double the number of chromosomes (tetraploidy). The cause of the aberrant genome-doubling remains unknown. Loss of DNA methylation is also frequent in cancer cells and has been found to be associated with aneuploidy. The mechanisms linking these alterations remain unclear. In this report, we describe the association between loss of methylation in a family of pericentromeric DNA sequences and sporadic genome-doubling in colorectal cancer cells cultured in vitro. These sequences are also hypomethylated in primary colorectal tumors, associated with inactivating mutations on genes of the main pathway controlling proper genome duplication. Our data suggest that the demethylation of these sequences might be associated with genome-doubling as early events in a subset of colorectal cancers, providing novel clues on the link between genome demethylation and aneuploidy in cancer.

**Abstract:**

Somatic DNA hypomethylation and aneuploidy are hallmarks of cancer, and there is evidence for a causal relationship between them in knockout mice but not in human cancer. The non-mobile pericentromeric repetitive elements SST1 are hypomethylated in about 17% of human colorectal cancers (CRC) with some 5–7% exhibiting strong age-independent demethylation. We studied the frequency of genome doubling, a common event in solid tumors linked to aneuploidy, in randomly selected single cell clones of near-diploid LS174T human CRC cells differing in their level of SST1 demethylation. Near-diploid LS174T cells underwent frequent genome-doubling events generating near-tetraploid clones with lower levels of SST1 methylation. In primary CRC, strong SST1 hypomethylation was significantly associated with global genomic hypomethylation and mutations in *TP53*. This work uncovers the association of the naturally occurring demethylation of the SST1 pericentromeric repeat with the onset of spontaneous tetraploidization in human CRC cells in culture and with *TP53* mutations in primary CRCs. Altogether, our findings provide further support for an oncogenic pathway linking somatic hypomethylation and genetic copy number alterations in a subset of human CRC.

## 1. Introduction

### 1.1. DNA Methylation Alterations in Cancer

DNA methylation is essential for the establishment and maintenance of cell-type-specific transcriptional profiles during cell differentiation, conferring cell type identity. It is also involved in suppressing the potentially harmful mobilization of endogenous transposable elements [1,2]. In humans, DNA methylation takes place almost exclusively in the cytosine residues within CpG dinucleotides. In differentiated somatic cells, most (70–90%) CpG sites are methylated, depending on the cell type [3,4]. DNA methylation is a dynamic process mediated by the activity of DNA methyltransferases and TET enzymes that are involved in the deposition and removal of DNA methylation, respectively [5]. During the lifespan of an individual, the level of genomic methylation in normal tissues declines at a rate proportional to their proliferative potential [6,7,8]. Notably, while the overall level of methylation declines with aging, some loci become more methylated [9]. In addition, environmental factors (exposure to pollutants, tobacco smoking, nutritional factors, among others) as well as the microbiota have been associated with global or site-specific DNA methylation alterations [10,11].

DNA methylation alterations are a common feature of cancer cells, with both gains and losses of methylation at different loci, associated with changes in chromatin structure and aberrant gene transcriptional activity at those locations [12]. The gain of methylation (hypermethylation), particularly on gene-promoter-associated CpG islands, is generally associated with a more compact chromatin conformation and transcriptional silencing, while the loss of methylation (hypomethylation) is generally associated with an open chromatin conformation and transcriptional activation of the neighboring genes [13,14]. The effect of DNA methylation alterations, however, is not limited to just the neighboring genes. Transcriptional dysregulation spanning large chromosomal regions (>1 Mb) containing both DNA-methylated and neighboring unmethylated genes that became coordinately suppressed by global changes in histone modification has been described in colorectal cancer (CRC) [15,16] and other solid tumors ([17] and references therein). The opposite phenomenon, i.e., regions with long range epigenetic activation, has also been reported [18]. Moreover, much of the methylation change in cancer involves regions exhibiting coordinated focal hypermethylation concentrated within regions of long-range (>100 kb) hypomethylation that are associated with late-replication domains of the genome, indicating a link between genome duplication and DNA methylation alterations [19,20,21]. Coordinated DNA methylation changes are not exclusively in *cis* genes but also occur in genes located in different chromosomes. One of the most widely studied examples is the coordinated CpG island promoter hypermethylation, referred to as the CpG island methylator phenotype (CIMP), first identified in CRC [22]. CIMP is associated with proximal CRCs with *BRAF* V600E mutation and microsatellite instability (MSI) caused by the hypermethyaltion of *hMLH1*. The underlying reason for the BRAF-CIMP-MSI association was found to be the RAF-MEK-ERK-induced phosphorylation of MAGF, a transcriptional repressor that upon phosphorylation binds the promoters of *hMLH1* and other CIMP genes and recruits a set of corepressors that includes the DNA methyltransferase *DNMT3B*, resulting in hypermethylation and transcriptional silencing [23]. This pathway is also involved in aberrant gene-promoter hypermethylation in mutant-*BRAF* melanomas [24]. The coordinated epigenetic silencing of tumor suppressor genes can be also mediated by *KRAS* oncogenic activation through a pathway involving the upregulation of ZNF304, a DNA-binding protein that recruits a corepressor complex that includes the DNA methyltransferase DNMT1 [25]. All these findings highlight the complexity of the mechanisms underlying the aberrant DNA methylation in cancer, including 3D genomic organization, altered signaling pathways, environmental factors and cell-cycle-associated alterations.

The genome-wide hypomethylation typically found in cancers [13] mainly reflects the loss of methylation of interspersed repetitive DNA elements that account for up 45% of our genome and over 90% of the methylated CpG sites [26,27,28]. Among these, LINE-1 elements have been widely employed as surrogate markers of global DNA methylation due to their high CpG content and abundance [29,30]. Genome-wide hypomethylation is associated with copy number alterations in human cancers [31,32] and leads to tumor formation in mice with knocked out methyltransferases, accompanied by chromosomal instability [33]. We previously found that global hypomethylation in gastric and colorectal cancers increased with patient age and correlated with genomic damage [34]. Based on these findings, a “wear and tear” model, linking aging to cancer development, was proposed. It postulated that changes in methylation—particularly hypomethylation—occur in normal stem cells or their descendants during aging because of the inevitable accumulation of errors during genome replication. When a minimum tolerable demethylation level is reached, or some particular sequences are affected, proper mitosis would be compromised, leading to chromosomal missegregation, transcriptional dysregulation and putatively contributing to oncogenesis [34]. 

### 1.2. Hypomethylation of SST1 Pericentromeric Repeats in Cancer

One of the *loci* that we found frequently hypomethylated in gastrointestinal tumors corresponded to pericentromeric repeated elements SST1 [34,35], also known as NBL2. These sequences were previously found to undergo hypomethylation in neuroblastoma [36] and methylation changes (both hypomethylation and hypermethylation) in different types of cancer [37]. SST1 sequences are also hypomethylated in cells from patients with immunodeficiency, centromeric region instability, facial anomalies syndrome (ICF) [38], associated with mutations in the DNMT3B DNA methyltransferase gene [39,40]. 

SST1 elements are 1.4kb long sequences moderately repeated in tandem clusters [41], mainly located near the centromeres of the short arm of acrocentric chromosomes 13, 14, 15 and 21. We identified that these sequences were mostly methylated in normal colonic mucosa samples, exhibiting an average of 88.7% ± 9.0% methylation in their CpG-rich 317bp internal region. Around 22% of the CRCs were hypomethylated (>5% lower methylation compared with matched normal tissue), including 7% of CRCs that exhibited strong hypomethylation of SST1 elements (≥10% demethylation compared with matched normal tissue), which we named “severe” hypomethylation. The majority of the CRCs (78%) exhibited no hypomethylation, with a few of them (around 5%) exhibiting moderate hypermethylation (>5% hypermethylation compared to matched normal tissue). Notably, no association between SST1 hypomethylation and patient age was found, and, in fact, strong hypomethylation (≥10% decrease in methylation) occurred in patients younger than the median age, suggesting the existence of some specific demethylation mechanism targeting SST1 sequences other than the age-dependent stochastic process proposed in our “wear and tear” model [34]. We found that SST1 hypomethylation associated with the down-regulation of *HELLS* (Helicase Lymphoid-Specific Enzyme) [35], a DNA-helicase that plays an essential role in the maintenance of proper levels of methylation of repetitive DNA elements [42]. SST1 hypomethylation associated with changes in the histone code of these sequences [35], and with their transcriptional expression in the form of a long non coding RNA (tumor-associated NBL2 transcript, TNBL) stable throughout the mitotic cycle, that formed a perinucleolar aggregate in the proximity of a subset of SST1 *loci* during interphase [43,44]. The transcription of SST1 sequences had been previously reported, albeit it was regarded as the product of run-through transcription rather than SST1 specific promoter activity [37]. Despite the evidence of the hypomethylation of SST1 sequences as a somatic alteration putatively contributing to CRC and other cancer types, its phenotypic effects in cancer cells are still unclear. 

### 1.3. The Role of Aneuploidy in Cancer

Cancer cells exhibit numerous types of genetic alterations, i.e., chromosomal rearrangements, amplifications, deletions, point mutations, etc. An abnormal number of chromosomes, or aneuploidy, is one of the earliest discovered and most prevalent characteristics of cancer cells. Around 90% of solid tumors exhibit some degree of aneuploidy. More than one century ago, Theodor Boveri proposed that aneuploidy could, in fact, promote malignant transformation [45,46]. In the 1970s, before the era of cellular oncogenes and tumor suppressor genes, there was a general support for this view [47]. Whether aneuploidy is the driving force of malignant transformation is still debated [48]. 

Aneuploidy is detrimental to the fitness of non-malignant cells during development or when introduced experimentally. However, it is tolerated by cancer cells, and, in some contexts, it might even contribute to their fitness [49]. Aneuploidy can be generated by different mechanisms. After chromosomal alignment in metaphase, sister chromatids are segregated during anaphase. In most cells, cytokinesis occurs simultaneously, causing the cytoplasm to divide into two daughter cells accommodating the newly segregated chromosomes [50]. The failure of these processes can affect genomic duplication fidelity. Furthermore, failures during cytokinesis can lead to complete genome duplication, or tetraploidy [51]. 

### 1.4. The Importance of Tetraploidy in Cancer

Genome-doubling and, consequently, tetraploidy is perhaps one of the most underestimated precursors of aneuploidy and chromosomal instability in human cancers [48]. Tetraploid cells are substantially more tolerant to deleterious mutations, since they harbor multiple copies of every gene, thus facilitating cell survival in a mutation-prone environment such as that associated with many types of cancers. In addition, tetraploidy may trigger chromosomal instability (CIN), probably due to centrosome abnormalities and duplicated chromosome mass, leading to aneuploidy [52,53]. There is experimental evidence directly linking tetraploidy with the onset of tumorigenesis in mice [53,54].

Analyses of chromosome copy number alterations of human cancers have shown that in the ontogenic genealogy of around one third of all solid tumors, a prior doubling event of the genome can be detected. These cancers have a post genome-doubling relative higher rate of chromosome gain and losses [55,56]. Tetraploid cell lines tolerate chromosomal segregation errors better than their diploid precursors [56,57]. While aneuploidy generates allelic imbalances that can be straightforwardly detected using molecular techniques, such as fingerprinting (reviewed in [58]), comparative genome hybridization arrays (aCGH) ([59], reviewed in [60]), SNP arrays [61] and even methylation arrays [62], tetraploidy evades detection using these techniques because the relative chromosomal content, allele balance and presumably methylation profiles remain unaltered.

### 1.5. TP53 and Aneuploidy in Cancer

The protein of Tumor Suppressor Gene P53 (*TP53*) monitors several aspects of DNA repair and cell division and contributes to suppressing tumor progression by hindering the division of cells with damaged DNA. Because of this, *TP53* has been named the “guardian of the genome” [63]. *TP53* mutations are very common in cancer [64] and according to TCGA data, they are the only cancer gene mutations significantly associated with CIN [65]. They also associate with tumors displaying evidence for genome duplication [66]. Moreover, it has been shown that wild type p53 blocks the growth of tetraploid cells, through G1 arrest [53,67].

In this work, we provide experimental evidence for a link between the demethylation of the pericentromeric SST1 repeats and the onset of tetraploidy in human CRC cells in culture and for the association of strong demethylation of SST1 with mutations in *TP53* in primary CRC. 

## 2. Materials and Methods

### 2.1. Cell Lines and Tumor Samples

Colorectal cancer cell lines LS174T (CL-188), HCT116 (CCL-247), DLD-1 (CCL-221) and ovarian cell line OV-90 (CRL-11732) were obtained from the American Type Culture Collection (ATCC). Primary CRCs with matched normal tissue from 148 patients were obtained as frozen specimens from the Cooperative Human Tissue Network (CHTN, https://www.chtn.org/). Informed consent was obtained and managed by the CHTN at the time of sample collection, following US regulations and the guidelines of the Declaration of Helsinki [68]. Approval to collect samples, extract DNA and perform analyses was obtained from the Institutional Review Board (IRB) of the Sanford-Burnham-Prebys (SBP) Medical Discovery Institute. Additional approval was obtained from the IRB of the IGTP to further analyze DNA samples in this study (approval code PI-18-252, 14 December 2018). 

The identity of all the cells used in this work, as well as their derived clones, was validated by STR analysis using AmpFLSTR™ Identifiler™ Plus PCR Amplification Kit (Cat n:4486467, ThermoFisher Scientific, Waltham, MA, USA) using the provided protocol. Sequencing results were analyzed using GeneMapper ID v3.2 (Applied Biosystems, Waltham, MA, USA) and analyzed with Expasy Cellosaurus tool CLASTR 1.4.4 [69]. Data are provided in Appendix A. All cell lines were confirmed to be mycoplasma-free by PCR analysis, prior to the initiation of the experiments and then regularly during the experimental timeframe. 

### 2.2. Cell Culture Conditions

LS174T, HCT116 and DLD-1 cell lines were cultured in DMEM:F12 medium supplemented with 10% fetal bovine serum (FBS), 2 mM L-glutamine, 1 mM sodium pyruvate (NaPyr) and Antibiotic-Antimycotic. Ovarian cell line OV-90 was cultured in ATCC recommended media, 1:1 MCDB 105 medium containing a final concentration of 1.5 g/L sodium bicarbonate and Medium 199 containing a final concentration of 2.2 g/L sodium bicarbonate supplemented with 15% FBS and antibiotic and antimycotic. All cells were cultured in ø100-mm culture plates, or in 24-well plates with coverslips, in a 37 °C incubator with 5% CO_2_. Cells were grown until reaching 80–90% confluency before the collection or passage.

### 2.3. Isolation of Single Cell-Derived LS174T Cell Clones

LS174T and OV-90 cell line subcloning was performed using the limiting dilution method [70], as we have already performed in previous works for the isolation of spontaneous frameshift mutations in MMR genes in CRC cells with microsatellite instability (MSI) and to study their phenotype consequences [71]. Cells were counted after the addition of Trypan Blue 0.4% 1:1 and diluted in complete medium to a concentration of 5 cells/mL (0.5 cells/100 µL). Moreover, 100 µL of this dilution were seeded into each well of 96 plates and incubated at 37 °C. Cell growth was monitored by microscopy. The occasional presence of more than a single growth focus during culture was used as indication of non-clonality, and those cultures were discarded.

### 2.4. Nuclei Size Measurement

Cell morphology was studied by fluorescence microscopy after phalloidin and DAPI staining (Phalloidin-iFluor 594 Reagent, ab176757—Abcam). Cells were cultured in 24-well plates with glass coverslips inside until they reached 70–80% confluence. Then, the culture medium was carefully removed and washed once with PBS. Cells were fixed by incubation with 3–4% formaldehyde in PBS at room temperature for 20 min. The fixation solution was aspirated, and fixed cells were washed 3 times with PBS. To increase permeability, cells were incubated in PBS + 0.1% Triton X-100 for 5 min and washed with PBS 3 times. Next, 100 µL of 1× Phalloidin conjugate (1 µL Phalloidin-iFluor 594 1000× in 1 mL PBS + 1% BSA) was added to each coverslip. They were incubated at room temperature for 60 min, protected from light. Finally, 100 µL of PBS + 1 µg/mL DAPI (Cat N.: D1306—Thermo Fisher Scientific, Waltham, MA, USA) were added to stain and visualize the nuclei. Lastly, 3 washes with PBS were performed to remove DAPI excess. The samples were dehydrated by immersion in 95% ethanol during 15 s and left to dry for 10 min in total darkness and mounted with ProLong Gold antifade reagent (Invitrogen, Thermo Fisher Scientific, Waltham, MA, USA). Samples were observed and photographed by fluorescence microscopy (LEICA DM1600B Microscope—LAS X Software). Fluorescence microscopy images (200×) were analyzed with ImageJ Software [72]. The area of at least 200 nuclei was quantified per sample. 

### 2.5. Karyotyping

Karyotypes were prepared from mitotic cells arrested in metaphase using colcemid, lysed in hypotonic solution (KCl 0.075M), fixed with Carnoy’s solution (methanol:glacial acetic acid, 3:1) and stained according to the G-banding pattern, following standard procedures. The analyses were conducted with cells of the parental cell lines and derived single-cell clones at different time points of the culture after single-cell clone isolation. Karyotypes with tetraploidy were true tetraploid and not mixtures of two diploid cells, as determined by the mitotic patterns under microscopy, as well as the sparse cell densities in the cultures treated with colcemid for mitotic arrest.

### 2.6. DNA and RNA Extraction

Genomic DNA from cell lines was isolated using Maxwell^®^ 16 Instrument and the Maxwell^®^ 16 DNA Purification Kit (Cat.# AS1020, Promega, Madison, WI, USA) following the protocol provided. DNA concentration was measured by NanoDrop^TM^-1000 UV/Vis Spectrophotometer (Thermo Fisher Scientific, Waltham, MA, USA) and visualized in an agarose gel. Total RNA from cell lines was isolated using Maxwell^®^ 16 Instrument and the Maxwell^®^ 16 LEV simplyRNA Cells Kit (Cat.# AS1270, Promega, Madison, WI, USA) using the protocol provided. RNA quality control and integrity was measured using Agilent 2100 Bioanalyzer (Agilent Technologies, Santa Clara, CA, USA). All samples had an RNA Integrity Number (RIN) above 9. 

### 2.7. LINE-1 MS-QPCR

LINE-1 MS-QPCR is a technique developed to estimate the ratio between unmethylated vs. methylated LINE-1 elements. This technique employs two sets of primers, one to amplify methylated (LINE1-mF and LINE1-mR) and the other unmethylated (LINE1-uF and LINE1-uR) LINE-1 molecules. Primer sequences are detailed in Appendix A. Two separate reactions were performed in duplicate for every sample. Conditions per reactions were: 5 µL of Roche MasterMix SYBR-Green 2X, 100 ng of bisulfite-transformed DNA, primers at 0.4 µM each, 10 µL final volume, 1 cycle of 95 °C 10′ followed by 40 cycles of 95°C 10″, 60 °C 10″ and 72 °C 12″, 1 cycle of 95 °C 5″, 60 °C 1′ and 97 °C. Quantification was made with LightCycler^®^ 480 Software using a relative quantification program with standard settings and calculating Cts by double derivative method. All measures were normalized vs. genomic DNA from DLD-1 cells. LINE-1 MS-QPCR provide the relative demethylation level of LINE-1 elements (LINE-1 RDL), i.e., an estimation of the ratio of demethylated vs. methylated LINE-1 sequences in the studied sample relative to that of DLD-1 genomic DNA. Therefore, higher RDL indicates higher demethylation, i.e., lower level of methylation. For the statistical analysis, LINE-1 RDL was log2 transformed to reduce the skewness of the distribution.

Albeit very similar in concept to LINE-1 MethyLight [30], LINE-1 MS-QPCR provides the ratio of unmethylated vs. methylated LINE-1 molecules from the same sample without relying on the amplification of any other type of repetitive elements, such as Alu elements. Thus, there is no effect of possible differences in Alu vs. LINE-1 abundance among samples due to germline or somatic copy number differences. Moreover, while LINE-1 MethyLight employs specific primers and internal probes for the methylated and unmethylated PCR reactions to increase specificity, LINE-1 MS-QPCR employs just specific primers but no internal fluorescent probes. We purposely selected this approach to increase the range of detected molecules in the unmethylated and in the methylated reactions. We estimated the specificity of the amplification by sequencing clones from the LINE-1 methylated reaction and from the LINE-1 unmethylated reaction (Appendix A). From the methylated LINE-1 reaction, the proportion of methylated CpG sites ranged from 58% and 100% (median value 91%). A total of 9 out of 10 clones from the unmethylated reaction showed complete demethylation, and 1 showed 22% methylation (2 CpG sites methylated and 7 unmethylated, on a sequence that exhibited 4 polymorphisms in the 13 interrogated CpG sites), for a median value of 0% methylation. Thus, LINE-1 MS-QPCR is not restricted to fully unmethylated or methylated sequences but instead provides the ratio of sequences with 0~22% methylation (unmethylated sequences) vs. 60–100% methylation (methylated sequences). 

### 2.8. SST1 MS-QPCR

SST1 MS-QPCR is similar to the LINE-1 MS-QPCR technique described above, but in this case providing the ratio between unmethylated SST1 vs. amplifiable methylation-independent SST1 sequences by using two sets of primers, one specific for unmethylated SST1 elements and the other amplifying SST1 elements irrespective of their methylation status (primer sequences are detailed in Appendix A). The primers for the methylation-independent SST1 sequences were SST1-F and SST1-R, which we employed for bisulfite sequencing in a previous publication [35]. These primers amplify a 317 bp internal region of the consensus SST1 sequence containing 27–28 CpG sites, some of them polymorphic, and including the NotI site that we originally found hypomethylated by MS-AFLP DNA fingerprinting [34]. Analyzing our bisulfite sequencing data from LS174T cells, we determined the CpG sites that better estimated the overall methylation level of this region (Appendix A). We designed the primer SST1-U-F to amplify SST1 sequences with unmethylated CpG3 (avoiding both CpG4, with low correlation with overall methylation, and CpG5, highly polymorphic) and primer SST1-U-R to amplify unmethylated CpG17 and CpG18 that exhibited the highest correlation with the overall methylation value (r = 0.77 and r = 0.79, respectively, Appendix A). Specificity of SST1-U-F and SST1-U-R primers was estimated by bisulfite sequencing (Appendix A). QPCR conditions and calculation of SST1 RDL were identical to LINE-1 MS-QPCR also using DLD-1 genomic DNA as reference sample (see above). For the statistical analysis, SST1 RDL was log2 transformed to reduce the skewness of the distribution.

SST1 MS-QPCR exhibited strong concordance with our previous results using bisulfite sequencing (correlation = 0.73, CI95% = [0.54,0.85], Pearson’s test *p* = 7.9 × 10^−8^, Appendix A). Considering bisulfite sequencing (originally used to classify cases into no-change/demethylated or strongly demethylated [35]) as gold standard, MS-QPCR was very accurate to detect strongly demethylated samples (max accuracy = 0.96, AUC = 0.79, *p* = 0.01). Samples with moderate changes in SST1 methylation were not detected by MS-QPCR, which was very likely because they do not have very low methylated SST1 molecules that are targeted by SST1-U-F and SST1-U-R primers. 

### 2.9. Bisulfite Sequencing

Genomic DNA was treated with sodium bisulfite using EZ DNA Methylation Kit (Cat.#D5001, Zymo Research, Irvine, CA, USA). Bisulfite treated samples were used as a template for PCR and QPCR amplification of SST1/LINE-1 sequences. 

A SST1-specific PCR was designed to cover a total 317 bp, including 27–28 CpGs including the NotI site of the band previously identified by MS-AFLP. The PCR Master Mix contained: 1× Buffer, 0.125 nM dNTPs, 0.25 units of Taq DNA Polymerase (Roche) and 0.4 mM of primers SST1-F and SST1-R (Appendix A). The PCR program consisted of 1 cycle at 95 °C for 5 min, followed by 35 cycles of denaturation at 95 °C for 15 s, annealing at 55 °C for 30 s and extension at 72 °C for 60 s, ending at 72 °C for 5 min to complete extension.

For bisulfite sequencing, 1 μL of SST1-F/R PCR product was cloned into pSC-A-amp/kan vector using Strataclone^TM^ PCR Cloning Kit (Cat.#240205, Agilent Technologies, Santa Clara, CA, USA) following manufacturer’s instructions and transformed into *E. coli* Statataclone SoloPack competent cells. Transformed cells were selected onto LB plates containing Ampicillin (50 μg/mL) and X-Gal (40 μg/mL). To identify bacterial colonies harboring the vector with the cloned PCR product, colony PCRs were performed using universal primers T3 and T7 [73]. Amplicon size was analyzed by gel electrophoresis and those reactions that rendered a correct size fragment were sequenced (GATC Services, Eurofins Genomics, Louisville, KY, USA) and analyzed with CLC Main Workbench 5.6.1 program. 

### 2.10. Illumina HM450K Methylation Arrays

Genomic DNA from tumors and matching normal tissues from 30 CRC patients were previously analyzed using Illumina HM450K arrays, following standard procedures [74]. The generated idat files were preprocessed and filtered using RnBeads [75], discarding data from X and Y chromosomes. Type-I vs. Type-II probe bias was corrected using the BMIQ method from the wateRmelon package [76]. 

### 2.11. Gene Expression Arrays and Gene Set Enrichment Analysis

RNA was extracted from cultured cells using Maxwell^®^ RSC simplyRNA Cells Kit on a Maxwell AS2000 equipment (Promega, Madison, WI, USA). Quantity and quality of RNA were verified on an Agilent 2100 Bioanalyzer. cDNA synthesis, labeling and hybridization on Clariom S Arrays from Thermo Fisher Scientific were performed by the Genomics Unit of Josep Carreras Leukaemia Research Institute. Raw data files were preprocessed using RMA (oligo R package). Gene set enrichment analysis was performed using GSEA, comparing with MSigDB hall mark gene set [77]. Full input dataset and output in HTML, specifying the running parameters, are provided as Appendix A.

### 2.12. Statistical Analyses

All statistical analyses were performed using R [78]. Normality was tested using Shapiro’s test. Variables normally distributed were analyzed using parametric tests (e.g., *t*-test, ANOVA). Variables not normally distributed were analyzed using Wilcox’s exact test. When appropriate, multiple hypothesis correction was performed using the false discovery rate (FDR) method implemented in p.adjust function. Comparison of coefficient of variation (*CV* = σ/μ) was performed using the asymptotic Feltz–Miller’s test [79]. 

## 3. Results

### 3.1. Isolation of LS174T Single Cell-Derived Clones

To test the hypothesis that the demethylation of the SST1 pericentromeric repetitive sequences could increase the probability of undergoing mitotic errors, we designed a strategy to study the effects of variations in the SST1 demethylation level in chromosomal changes in isogenic cells. Among 14 CRC cell lines, LST174T cells exhibited the lowest average level (55 ± 25%) and the highest range (19–88%) of SST1 methylation [35]. This suggested that it would be feasible to isolate single cell clones with different levels of SST1 methylation. We hypothesized that SST1 demethylation could be accompanied by an increased rate of chromosomal alterations, observable after long-term growth. The approach was facilitated because LS174T is near-diploid (45 X, after the loss of one of the X chromosomes), with no reported chromosomal instability. It exhibits MSI due to the hypermethylation of *hMLH1* but not CIMP [80]. We generated a collection of 30 single-cell-derived clones using the limited dilution method, from which we randomly chose 14 (C1 to C14). The identity of these individual clones as well as the original LS174T cells was confirmed by short tandem repeat (STR) analysis (Appendix A). 

### 3.2. LS174T Is a Mixture of Normal-Sized and Large-Nucleus Cells with Different Ploidy

We intended to subject these LS174T single-cell clones to long term culture, but clear differences in cell size among them were observed right after their initial expansion. The cell staining with phalloidin and DAPI revealed that cells from clones C5–C8 had substantially larger nuclei in interphase (Figure 1A). The clustering of the clones based on their nuclei area yielded two distinct groups: a group comprising clones C1–C4 and C9–C14, with a nuclei area of 72.5 ± 26.2 μm^2^, and a group comprising clones C5–C8, with a nuclei area of 132.1 ± 50 μm^2^ (Figure 1B). The nuclei area of the parental cell line was 89.7 ± 37.8 μm^2^ but exhibited a trimodal distribution with peaks at approximately 65, 115 and 180 μm^2^ (Appendix A) that could not be explained just by the coexistence of cells in G0/G1 vs. cells in S/G2. The doubling of a nucleus’s volume corresponds to a 2^2/3^ ≈ 1.6 times increase in its area. Thus, nuclei with areas of 115 μm^2^ would correspond to roughly two times the volume of nuclei with areas of 65 μm^2^ and nuclei with areas of 180 μm^2^ over four times that. To estimate the proportion of cells with large nuclei, a threshold of 115 μm^2^ was set based on the nuclei distribution of the parental LS174T cells. About 22% of the LS174T cells had a nucleus larger than 115 μm^2^. Over 50% of the cells in clones C5–C8 had a nucleus larger than that. Clones C1–C4, C10–C11 and C13–C14 had less than 10% of nuclei over 115 μm^2^. Two LS174T-derived clones classified as small according to their average nuclei area exhibited a higher proportion of nuclei larger than 115 μm^2^, i.e., C9 (11%) and C12 (15%) (Figure 1C). Flow cytometry analysis indicated that clones C5-C8 had approximately two times greater genomic content than clones C1–C4 and C9–C14 (Appendix A). 

Karyotyping revealed that parental LS174T was a mixture of near-diploid (45,X) and near-tetraploid (90,XX) cells, with the sporadic appearance of 46,X cells with trisomy of chromosome 7 (Figure 2A). In agreement with the nuclei size and cytometry results, clones C5, C6, C7 and C8 were composed by near-tetraploid cells (90,XX). Clones C1, C3, C4, C10 and C11 exhibited almost exclusively near-diploid nuclei (45,X). Clones C2, C13 and C14 exhibited cells with 46,X and trisomy of chromosome 7 (Figure 2B). Clones C9 and C12 were a mixture of near-diploid and near-tetraploid cells. Our interpretation was that some cells of C9 and C12 underwent spontaneous tetraploidization during the culturing time after isolation, indicating that genome duplication is a dynamic process in LS174T cells and not the result of a single past event. However, it was also possible that clones C9 and C12 were derived from two (or more) cells with different ploidy and substantially different proliferation rates favoring a larger proportion of near-diploid cells after several rounds of cell division. To investigate this possibility, we selected clones C8 and C11 as representative of pure near-tetraploid and near-diploid cells, respectively, and determined their karyotype after 60 days of culture, equivalent to over 40 cell divisions (Appendix A). Clone C8 cells exhibited an almost stable near-tetraploid karyotype. At time 0, 100% of the metaphases were near-tetraploid (97 out of 97). After 60 days of culture, we found 3 mitotic cells out of 72 with n > 90 chromosomes (Figure 2C). No near-diploid cells were found at any time point in this clone. On the other hand, clone C11 cells exhibited mostly near-diploid karyotype at time 0, with only 1 near-tetraploid metaphase out of the 95 mitoses analyzed (1%), but we found 10 out of 75 mitoses (13%) with near-tetraploid metaphases after 60 days of culture (*p* = 0.003, Fisher’s exact test) (Figure 2C). Since the individual cells analyzed for their karyotype originated from a clone that was near-diploid and the mitoses were typical of single cells and not mixtures of different cells (Figure 2D), we concluded that these near-tetraploid cells originated from near-diploid parent cells. Modeling the generation of near-tetraploid cells in clone C11, we estimated that approximately 3% of the mitosis resulted in spontaneous tetraploidization (Appendix A). 

### 3.3. Analysis of Other Cancer Cell Lines

To extend these observations, we analyzed the degree of anisonucleosis (variability in cell nuclei size) by calculating the coefficient of variation (CV) of the nuclei area as a proxy for the genomic content in HCT116 and DLD-1 CRC cell lines. The level of anisonucleosis in LS174T (CV = 42%) was higher than in HCT116 (CV = 33%, asymptotic test *p* = 4.2 × 10^−7^) and DLD-1 (CV = 38%, asymptotic test *p* = 0.036), despite the three cell lines being MSI and originally described as mainly diploid or near-diploid. Moreover, the proportion of cells with large nuclei (over three times the modal nucleus volume, where the modal nucleus volume was assumed to correspond to diploid or near-diploid cells in G0/G1) was significantly higher in LS174T (12.7%) than in DLD-1 (7%, Fisher’s test *p* = 0.004) and HCT116 (1%, Fisher’s test *p* = 6.3 × 10^−16^) (Appendix A). The higher incidence of polyploidy in our cultures of LS174T cells (12.7%) than in the first report describing this cell line (6.1%) [81] is likely due to the accumulation of near-tetraploid cells during successive culture passages. Of note, near-tetraploid cells have been also observed in LS-180, the LS174T parental cell line [81], albeit at a lower rate (2.2%, according to the ATCC). Due to the absence of significant SST1 demethylation in DLD-1 and HCT116 cell lines (Appendix A), a similar study to isolate single cell clones with substantially different levels of SST1 methylation was not feasible.

We generated 15 single-cell clones from the ovarian cancer cell line OV-90, which displayed a wide range and a low average of SST1 methylation (50 ± 20%) in our previously published study [35]. OV-90 was originally described as mainly diploid, albeit displaying complex karyotypic changes involving chromosomal rearrangements, translocations, deletions and duplications [82]. In contrast with LS174T, OV-90 is microsatellite stable (MSS) and harbors a mutant *TP53* gene. The single-cell clones of OV-90 showed aberrant and very variable chromosomal content with an overall distribution of hypertriploidy–hypopentaploidy, with numerous multipolar spindle mitoses indicative of active chromosomal instability (Appendix A). In addition to chromosomes with two and three copies, the karyotypes presented several chromosomes with four copies (Appendix A). These results are consistent with an ancestral tetraploid cell lineage. All 15 subclones exhibited very high chromosomal instability and ploidy variability, and no distinctive groups based on their nuclei area or ploidy could be observed in the time frame of the cultures (data not shown). 

### 3.4. Chromosomal Content Correlates with SST1 Methylation Level in LS174T Cells

We analyzed the methylation level of the SST1 sequences in the 14 LS174T-derived clones, as well as the LS174T parental cell line, using bisulfite sequencing (see Section 2.9). The methylation of individual SST1 elements within each single-cell derived clone was highly variable (Figure 3A). Only 10.7% of the methylation variance was explained by the differences among clones, while 89.2% reflected intra-clonal variability (ANOVA analysis). Despite this variability, the near-tetraploid clones (C5–C8) exhibited lower average methylation than the near-diploid clones (30.1 ± 2.8% vs. 43.6 ± 4.0%, *t*-test *p* = 1.0 × 10^−4^). The clones with mixed populations of near-diploid and near-tetraploid cells (C9 and C12) also exhibited low levels of methylation (30.5 ± 3.8%). These results indicated a significant inverse correlation between SST1 methylation and tetraploidization in LS174T cells (r^2^ = 0.53, *p* = 0.002, Figure 3B). 

### 3.5. Transcriptional Profile Differences between Near-Diploid and Near-Tetraploid LS174T Cells

We compared the transcriptional profile of near-tetraploid (clones C5–C8) vs. near-diploid (clones C3, C10, C11 and C14) LS174T cells using Affymetrix Clariom^TM^ S arrays (Appendix A). The near-diploid and near-tetraploid LS174T subclones exhibited similar transcriptional profiles (Appendix A). A gene set enrichment analysis (GSEA) [77] revealed that the expressions of E2F targets, MYC targets and G2M checkpoint genes were significatively enriched in the near-tetraploid cells (Appendix A), consistent with the fact that the control of the cell cycle is altered in these cells [83]. 

### 3.6. SST1 Demethylation Associates with LINE-1 Demethylation in Primary CRCs

We then analyzed SST1 and LINE-1 methylation in 148 primary CRC tumors and their matching normal tissues using MS-QPCR, a technique that provides an estimation of the proportion of unmethylated target sequences, a value termed as the relative demethylation level (RDL, see methods). Clinicopathological, mutational and DNA methylation data from these 148 CRC are shown in Appendix A. To benchmark MS-QPCR, 40 of these cases were chosen among those previously analyzed by bisulfite sequencing. We obtained very concordant results between both techniques (r = 0.73, *p* = 8 × 10^−8^, Appendix A). 

SST1 RDL in tumors and matching normal tissues strongly correlated (r = 0.64, *p* = 3.9 × 10^−18^, Figure 4A). Most tumors exhibited a degree of SST1 methylation very similar to that of their matching normal tissues, and the average methylation level was similar in both groups (paired *t*-test *p* = 0.07, Figure 4B). Seven tumors, however, deviated from this trend and exhibited SST1 RDL values higher than the 95% confidence interval of the regression model prediction. These tumors were classified as SST1 strongly demethylated (Figure 4A,B). The methylation of LINE-1 exhibited a very different behavior. We found no significant correlation between the tumor tissue and the matching normal tissues (r = −0.06, *p* = 0.56, Figure 4C), and most tumors exhibited lower methylation than their matching normal tissues (paired *t*-test *p* = 6 × 10^−23^, Figure 4D). 

We defined the somatic demethylation value (ΔRDL) as the difference between RDL from the tumors and their matching normal samples, after log2 transformation. Thus, ΔRDL = 0 indicates the same proportion of demethylated SST1 molecules in the tumor and the matching normal sample, ΔRDL = 1 indicates that the proportion of demethylated SST1 molecules is two times larger in the tumor than in the normal, etc. All seven outlier cases classified as SST1 strongly demethylated exhibited SST1 ΔRDL > 3. 

SST1 somatic demethylation correlated with LINE-1 somatic demethylation (r = 0.51, CI95% = [0.36–0.64], *p* = 1.5 × 10^−8^, Figure 4E). This correlation was highly influenced by the few cases with strong SST1 somatic demethylation, all of them exhibiting high levels of LINE-1 somatic demethylation. Excluding these cases, the correlation between SST1 ΔRDL and LINE-1 ΔRDL was much lower, albeit still positive (r = 0.2, CI95% = [0.009–0.39]).

The genome-wide methylation profiling of 30 CRCs and their matching normal samples with Illumina HM450K arrays showed that LINE-1 somatic demethylation correlated with genome-wide demethylation (r = 0.68, *p* = 4.4 × 10^−5^, Appendix A). When considering only the CpG sites not associated with CpG islands (those located > 2000 bp apart from their closest CpG island), the correlation remained high r = 0.69, *p* = 3.6 × 10^−5^ (Appendix A). This association was expected since LINE-1 methylation is widely considered a marker of global genome methylation. SST1 demethylation, on the other hand, did not correlate with genome-wide demethylation regardless of the analyzed subset of CpG sites (Appendix A). 

### 3.7. SST1 Strong Demethylation Is Associated with TP53 Mutations in CRC

CRCs were classified into three categories based on their SST1 ΔRDL: tumors with strong (ΔRDL > 3, n = 7, 4.7%), moderate (1 < ΔRDL < 3, n = 18, 12.2%) and without (ΔRDL < 1, n = 123, 83.1%) somatic demethylation. Associations between SST1 somatic demethylation and clinicopathological characteristics of the 148 CRCs are shown in Table 1. SST1 somatic demethylation correlated with *TP53* mutations in both comparisons (moderate or strong demethylation vs. rest, *p* = 0.0037, and strong demethylation vs. rest, *p* = 0.012, Fisher’s test). To reduce ambiguity around the RDL measurement error influencing sample categorization, we also compared tumors without demethylation (ΔRDL < 1) vs. tumors with strong demethylation (ΔRDL > 3) and confirmed that the association remained statistically significant (*p* = 0.0037) despite the lower number of samples. Strong demethylation exhibited a borderline significant correlation with wild-type *KRAS* (*p* = 0.048). We also studied the association of the clinical and mutational characteristics with SST1 ΔRDL as a continuous variable without performing a categorical classification of the cases (Appendix A). There was a significant correlation of SST1 ΔRDL with *TP53* mutations (*p* = 0.0015), with *KRAS* wild-type tumors (*p* = 0.017) and with African Americans (*p* = 0.007). In a multivariate regression analysis, only *TP53* mutations remained statistically significant (*p* = 0.026). 

## 4. Discussion

The original aim of this work was to explore the link between the demethylation of SST1 elements and chromosomal instability in human CRC cells. The approach employed LS174T, an MSI and mainly near-diploid CRC cell line with low average level and high variability (55 ± 25%) of SST1 methylation [35]. We found that right after isolation, some LS174T single cell-derived clones had a larger average cell and nuclei size (clones C5 to C8, Figure 1) with near-tetraploid karyotype (Figure 2). In agreement with our initial hypothesis, near-tetraploidy in LS174T clones was strongly associated with lower levels of methylation of SST1 elements (r^2^ = 0.53, *p* = 0.002, Figure 3). 

Near-tetraploid cells were also found in originally near-diploid clones after very few cell divisions (clones C9 and C12, Figure 2B). Longer culturing time (~40 cell divisions) of pure near-diploid cells (clone C11) generated near-tetraploid cells, reaching around 13% of the final population (Figure 2C). This showed that spontaneous tetraploidization in LS174T was not the result of a single past event, but the result of an intrinsic and active chromosomal segregation defect taking place with an estimated rate of 1 in 37 cells entering mitosis (around 3% of the mitoses, Appendix A). 

Near-tetraploid cells were stable and did not revert to the near-diploid status, at least within the studied timeframe. They, however, grew at approximately 80% the proliferation rate of the near-diploid cells (Appendix A), possibly due to the extra time required to duplicate a much larger genome. A mathematical model of the proportion of near-diploid and near-tetraploid cells in a mixed cell population, considering their different growth rates and the rate of spontaneous tetraploidization predicted that the proportion of near-tetraploid cells would stabilize at around 13% of the population (Appendix A). This prediction is in line with the proportion of cells with nuclei size above 3× de modal volume in the parental LS174T cell population (12.8%, Appendix A).

In primary CRCs, strong somatic demethylation of SST1 is associated with TP53 mutations and LINE-1 demethylation (Figure 4F). The link between SST1 hypomethylation and spontaneous tetraploidization might explain its association with *TP53* mutations, where the inactivation of *TP53* would be necessary to evade tetraploidy-triggered cell cycle arrest [53,67,84]. While LS174 cells are *TP53* wild-type, they harbor a homozygous frameshift mutation in *BAX* and also in other pro-apoptotic genes. *BAX* is a common MSI target because it contains a mutational hotspot (G)_8_ mononucleotide tract in its coding sequence [85,86]. The inactivation of *BAX* facilitates the survival of tetraploid cells at least as efficiently as the p53 or p21 knockouts [87]. Our experiments indicate that upon tetraploidization, LS174T cells activated the G2M checkpoint genes (Appendix A), but nevertheless, they continued to proliferate and did not enter apoptosis or cell cycle arrest. 

Our findings are summarized in the model presented in Figure 5. In this model, SST1 demethylation takes place in a globally demethylated genome, resulting in an increased probability of spontaneous tetraploidization that in turn triggers p38-p53-p21-mediated cell cycle arrest. Therefore, cells that undergo tetraploidization would require the impairment of the p38-p53-p21 pathway either by mutations in *TP53* (in MSS tumors) or by inactivation of essential downstream effectors such as *BAX* (in MSI tumors) to evade cell cycle arrest [67,84].

The cause of drastic SST1 demethylation in some tumors remains to be elucidated. Strong SST1 somatic demethylation was associated with high levels of LINE-1 hypomethylation in primary CRCs (Figure 4F), suggesting some common underlying mechanisms. This association was also observed in LS174T cells that exhibited much lower methylation in both SST1 and LINE-1 elements compared to DLD1 and HCT116 (Appendix A). However, there were fundamental differences in the hypomethylation affecting SST1 and LINE-1 sequences: While SST1 methylation in tumor samples mainly reflected the methylation of their matching normal tissue, LINE-1 elements were generally somatically hypomethylated in tumors (Figure 4). Therefore, the events that lead to strong demethylation in SST1 elements, while they may occur in a genetic background of global genomic demethylation, likely represent a different mechanism. We do not consider mutant p53 to directly underlie the strong demethylation of SST1 elements because the spectrum of *TP53* mutations in the cases with strong demethylation is not atypical, with common single point mutations in typical codons (Appendix A). In this context, we previously reported that the demethylation of SST1 was associated with the downregulation of *HELLS*, a protein that has been denominated the epigenetic guardian of the repetitive elements [42]. 

A limitation of our study is that, due to their repetitive nature, we could not determine the precise chromosomal location of the SST1 elements that undergo somatic demethylation. We cannot rule out that demethylation of some of these elements, but no others, might have different phenotypic effects. Of note, LS174T subclones did not show gross alterations of any of the acrocentric chromosomes, where SST1 elements are mainly located, regardless of their average SST1 methylation level or ploidy. Preliminary analysis by CGH arrays of a small subset of CRCs with strong SST1 demethylation did not reveal a higher incidence of alterations in acrocentric chromosomes. Thus, we have no evidence for a regional effect caused by SST1 demethylation in CRC. In line with this observation, lymphocytes from ICF patients, which also exhibit hypomethylation of SST1 elements [38], do not typically show structural alterations on the acrocentric chromosomes [88]. ICF cells are deficient for DNMT3B, leading to the extensive hypomethylation of repetitive elements other than STT1, i.e., LINE-1, Alu, Sat2, Sat3 and Yhq, aberrant decondensation of centromere-adjacent heterochromatin and structural alterations in chromosomes 1, 16 and sometimes 9 [30,88]. Still, tetraploidization is not typical of ICF cells perhaps because, unlike CRCs with strong SST1 demethylation, ICF cells are p38-p53-p21-competent.

The mechanisms linking SST1 demethylation with whole genome duplication also remain unresolved. The absence of structural alterations on the acrocentric chromosomes in LS174T cells suggests that regional chromatin decondensation does not play a major role. We recently reported that the demethylation of SST1 was associated with the expression of *TNBL*, a long non-coding RNA originating from these sequences and stable throughout the mitotic cycle, that formed a perinucleolar aggregate with RNA-binding proteins [43]. The function of this lncRNA, or its putative effect on genome duplication fidelity is still unknown. While SST1 demethylation could be linked to whole genome-doubling through a hypothetical negative effect on the cell cycle control, it is also plausible that it is not per se the direct cause of tetraploidization but a surrogate marker for the underlying epigenetic defect that leads to the genetic alteration.

Unfortunately, we cannot compare the data from our cohort with that of publicly available cohorts where whole-genome-doubling has been inferred [56], for instance, the TCGA COAD and READ datasets, because those samples have not been analyzed for SST1 methylation. The HM450K methylation arrays employed to profile methylation on TCGA samples do not cover these repetitive elements. Thus, the validation of our results has to be performed by analyzing additional fresh samples. SNP array or point-mutation-based methods that can infer tetraploid lineage in mixed cell populations may return definitive evidence [61,89]. 

Despite these limitations, this report provides important clues relative to the role of somatic DNA demethylation in human cancer. The hypothesized causal relationship between the demethylation of SST1 pericentromeric repetitive elements and genome duplication observed in LS174T cells is supported by its association with mutational inactivation of p53 or pro-apoptotic proteins in a subset of human primary CRCs. The correlation between global demethylation and copy number alterations has been reported by us and many other researchers [27,34,90]. There is also solid mechanistic evidence for the oncogenic effect of genome-wide demethylation in CRC cell lines, as well as in mice, in which the DNA methyltransferases (DNMTs) have been genetically disrupted [31,33,91]. Lastly, the association between LINE-1 hypomethylation and hyperploidy has been described in ovarian cancers [92]. However, to our knowledge, we present the first evidence linking naturally occurring DNA demethylation in human CRC cells with the onset of tetraploidy, putatively involved in the early steps of tumorigenesis in a subset of CRCs [52,93].

## 5. Conclusions

LS174T cells undergo spontaneous tetraploidization in vitro, associated with demethylation of SST1 repetitive elements. To the best of our knowledge, this is the first evidence linking naturally occurring DNA demethylation in human CRC cells with the onset of tetraploidy, which is likely involved in the early steps of tumorigenesis.

SST1 methylation in primary tumors reflects, in general, the methylation found in their matched normal tissues. SST1 demethylation occurs in around 17% of colorectal tumors (12% with moderate demethylation and 5% with strong demethylation), associated with mutations in *TP53*.

## Figures and Tables

**Figure 1 cancers-13-05353-f001:**
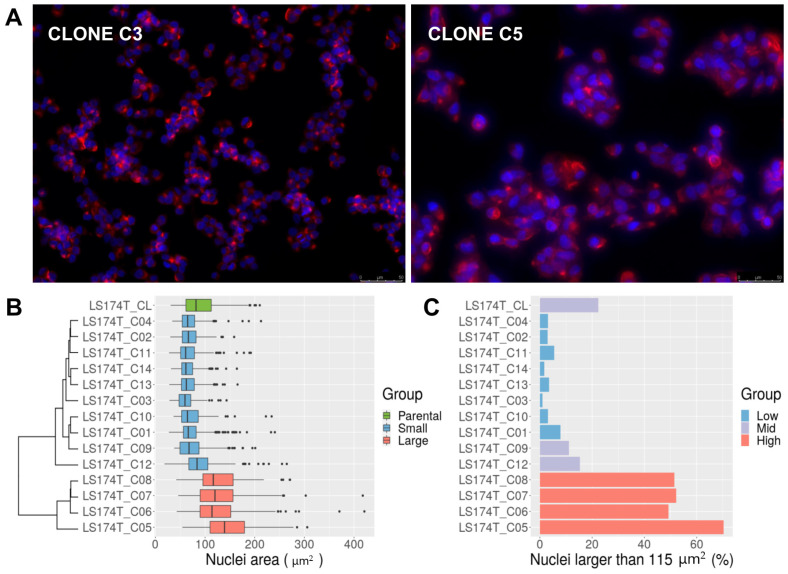
(**A**) Morphological analysis of LS174T clones. The figure shows a small cell size clone (C3) and a large cell size clone (C5) stained with phalloidin (red, cytoskeleton) and DAPI (blue, nuclei). Pictures were taken at 200× magnification. Scale bar 50 μm is shown in the bottom right corner of the pictures. (**B**) Nuclei area of LS174T and derived clones, measured by microscopy after DAPI staining. The parental cell line (CL) in green. Clones are colored according to the average nuclei size into small (blue) and large (red) and classified by unsupervised clustering based on the difference of their average nuclei size (dendrogram on the left). (**C**) Percentage of nuclei with areas larger than 115 μm^2^ in LS174T and derived clones. Most of the clones exhibited a low proportion (less than 10%, low group, blue) of nuclei above 115 μm^2^. The parental cell line (CL) and clones C9 and C12 exhibited a larger proportion (10~25%, mid group, in purple). Clones C5–C8 exhibited around 50% of nuclei larger than that value (high group, in red).

**Figure 2 cancers-13-05353-f002:**
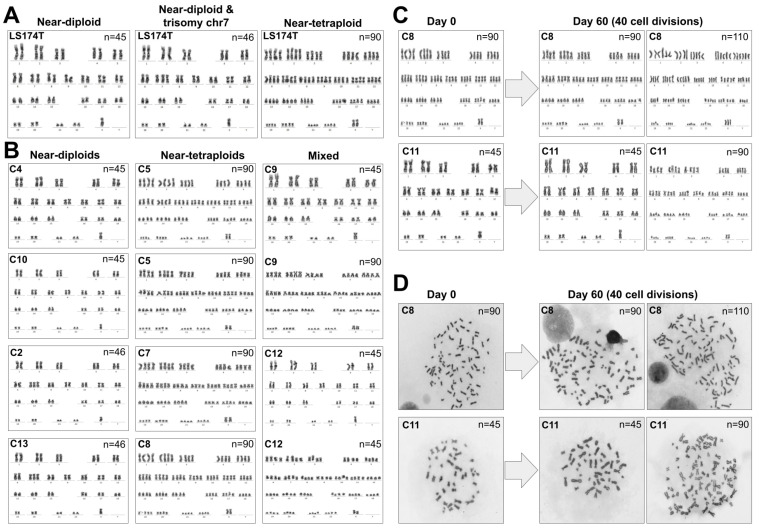
(**A**) Karyotypes of the parental cell line LS174T exhibited a mixture of near-diploid (n = 45, left), near-diploid with trisomy of chromosome 7 (n = 46, middle) and near-tetraploid cells (n = 90, right). (**B**) Left, karyotypes of near-diploid clones C4 and C10 (n = 45) and near-diploid clones C2 and C13 with trisomy of chr7 (n = 46); middle, near-tetraploid clones C5, C6, C7 and C8; right, clones with mixture of karyotypes C9 and C12. (**C**) Karyotypes of LS174T clones C8 and C11. At time 0 (left), C8 cells were pure near-tetraploids, and C11 cells were almost pure near-diploids. After 60 days of culture (right), C8 cells remained mostly near-tetraploid with the occasional appearance of higher ploidy cells, while C11 cells exhibited 13.3% of near-tetraploid cells. (**D**) Metaphases of the karyotypes presented in panel C.

**Figure 3 cancers-13-05353-f003:**
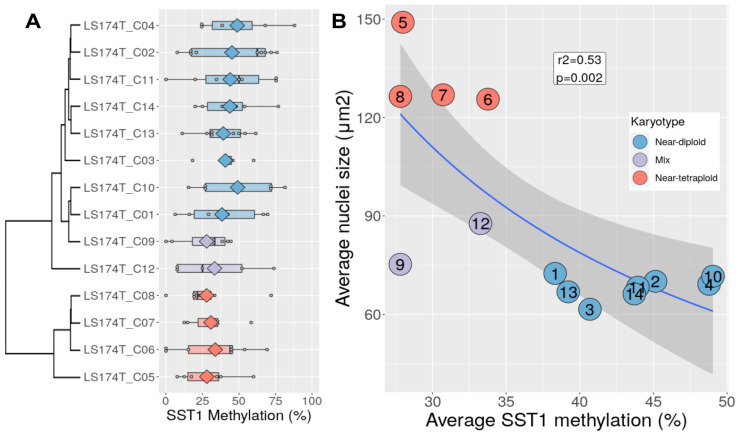
(**A**) SST1 methylation levels measured by bisulfite sequencing in LS174T-derived single cell clones. Clones are ordered according to their nuclei size-based clustering (see Figure 1) and colored according to their karyotype. Every grey dot represents the average methylation of an individually sequenced SST1 molecule. Diamonds indicate the average methylation level of all SST1 the molecules sequenced in every clone. (**B**) SST1 average methylation (*x*-axis) vs. average nuclei size (*y*-axis) in LS174T-derived single-cell clones. Every dot represents the value of a LS174T-derived cell clone, colored according to its karyotype. The regression line (nuclei area ~ 1/methylation) and its 95% confidence interval are depicted in blue and dark grey, respectively. The inverse correlation between nuclei area and methylation was r = 0.73, r^2^ = 0.53, *p* = 0.002.

**Figure 4 cancers-13-05353-f004:**
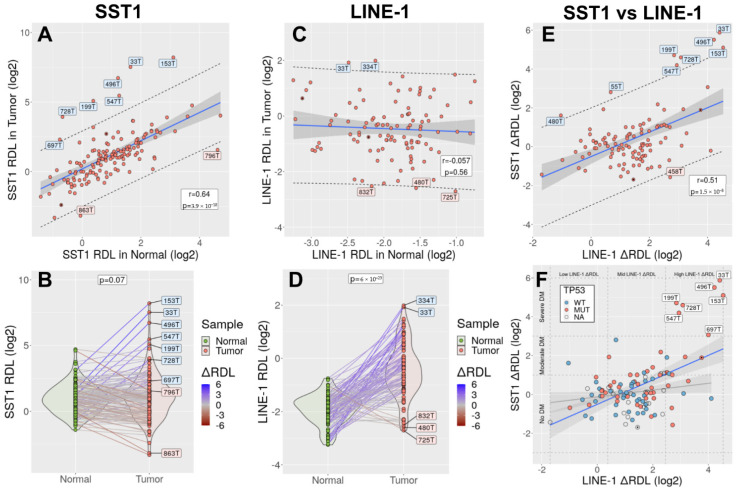
(**A**) Correlation between SST1 RDL in 148 CRC tumors (*y*-axis) and matched normal tissues (*x*-axis). Every dot represents a tumor. In blue, the regression line. Shaded in grey, the 95% CI of the slope. The dashed lines indicate the 95% CI of the values predicted by the regression model. Labeled in blue and red, tumors with SST1 RDL above (strongly demethylated) and below the predicted upper limit of the 95% CI, respectively. Two metastases are indicated with an internal black dot. (**B**) SST1 RDL values in normal tissues (green) and tumors (red). Tumor and matched normal samples are connected by solid lines, colored according to their difference in SST1 RDL (somatic demethylation, ΔRDL). Tumors with SST1 RDL values outside the 95% CI of the regression model prediction (shown in panel A) are labeled. (**C**) Correlation between LINE-1 RDL in 107 tumors and matched normal tissues. Symbols as in panel A. (**D**) LINE-1 RDL values in normal tissues and tumors. Symbols as in panel B. (**E**) Correlation between SST1 ΔRDL and LINE-1 ΔRDL. Symbols as in panels A and C. (**F**) Correlation between LINE-1 and SST1 somatic demethylation in 107 CRCs. Tumors are colored according to their TP53 mutational status (WT: wild type; MUT: mutant; NA: not analyzed). The graph is divided with dashed lines in nine areas according to the levels of LINE-1 ∆RDL (horizontal axis, divided in low, mid and high demethylated) and SST1 ∆RDL (vertical axis, divided into no demethylated, moderately demethylated and strongly demethylated). Two regression models are depicted, one with all the samples (blue line) and the other excluding the SST1 strongly demethylated cases (grey line). The dark grey areas indicate the 95% CI of the slope.

**Figure 5 cancers-13-05353-f005:**
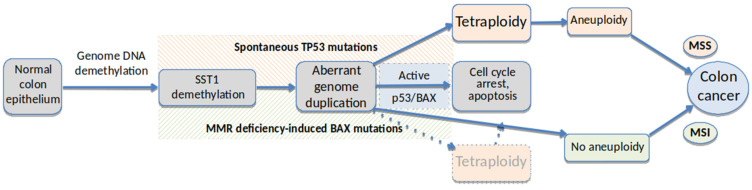
Model for a link between SST1 somatic demethylation, tetraploidy and TP53 and BAX mutations in CRC. In a genomic background of somatic DNA demethylation, strong demethylation of SST1 pericentromeric elements leads to genome duplication and consequent tetraploidy in CRC precursor cells. When cell cycle arrest or apoptosis networks are functional, the resultant tetraploid cells do not proliferate. When TP53 mutations in MSS CRC spontaneously occur before or during the presumably gradual demethylation of these (or other) sequences, the tetraploid cells survive initiating a pathway for aneuploidy and eventually CRC. In a less frequent MSI pathway for CRC, the tetraploid cells can evade apoptosis if there are somatic mutations in BAX and/or other pro-apoptotic proteins. In MSI CRC, the SST1 demethylation-associated tetraploidy is not essential for tumor progression, and the MSI mutator phenotype may lead to a pseudo-diploid tumor.

**Table 1 cancers-13-05353-t001:** Association of SST1 demethylation with CRC genotype and phenotype.

Parameter	Not Demethylated SST1 ΔRDL < 1.0	Moderately Demethylated SST1 ΔRDL > 1.0 & < 3.0	Strongly Demethylated SST1 ΔRDL > 3.0	Fisher’s Test *p*-Value
Gender	Women (n = 59)	Women (n = 6)	Women (n = 2)	*p*1 = 0.19
	Men (n = 64)	Men (n = 12)	Men (n = 5)	*p*2 = 0.46
Age	<66 y/o (n = 57)	<66 y/o (n = 10)	<66 y/o (n = 5)	*p*1 = 0.27
	>66 y/o (n = 66)	>66 y/o (n = 8)	>66 y/o (n = 2)	*p*2 = 0.27
Ethnic	Caucasian (n = 94)	Caucasian (n = 9)	Caucasian (n = 3)	*p*1 = 0.05
	Afr.Am. (n = 18)	Afr.Am. (n = 6)	Afr.Am. (n = 1)	*p*2 = 0.58
Tumor Location	Proximal (n = 69)	Proximal (n = 10)	Proximal (n = 2)	*p*1 = 0.38
	Distal (n = 50)	Distal (n = 8)	Distal (n = 5)	*p*2 = 0.24
Dukes’ stage	IS-A-B (n = 52)	IS-A-B (n = 9)	IS-A-B (n = 4)	*p*1 = 0.39
	C-D-M (n = 71)	C-D-M (n = 9)	C-D-M (n = 3)	*p*2 = 0.70
MSI status	MSS (n = 102)	MSS (n = 17)	MSS (n = 6)	*p*1 = 0.37
	MSI (n = 21)	MSI (n = 1)	MSI (n = 1)	*p*2 = 1.00
*TP53*	WT (n = 44)	WT (n = 5)	WT (n = 0)	*p*1 = 0.0037 **
	MUT (n = 32)	MUT (n = 11)	MUT (n = 7)	*p*2 = 0.012 *
*KRAS*	WT (n = 75)	WT (n = 12)	WT (n = 7)	*p*1 = 0.18
	MUT (n = 48)	MUT (n = 6)	MUT (n = 0)	*p*2 = 0.048 *
*BRAF*	WT (n = 112)	WT (n = 16)	WT (n = 4)	*p*1 = 0.25
	MUT (n = 10)	MUT (n = 2)	MUT (n = 2)	*p*2 = 0.10

*p*1: Fisher’s test *p*-value comparing not-demethylated tumors vs. the rest (demethylated and strongly demethylated). *p*2: Fisher’s test *p*-value comparing strongly demethylated tumors vs. the rest. *: *p*-value < 0.05. **: *p*-value < 0.01. In parenthesis, the number of informative cases in each category. In Ethnic, Afr.Am.: African American. In Duke’s stage, tumors were grouped into in situ (IS, n = 1), Duke’s A (n = 7) and Duke’s B (B, n = 54) vs. Duke’s C (n = 58), Duke’s D (n = 23) and metastases (M, n = 2). WT: wild-type. MUT: mutant. Some of the total number of samples do not match the total 148 cases sample due to incomplete information.

## Data Availability

All data is included in the manuscript as Appendix A. Original cell lines are publicly available at ATCC. Derived clones from our collection are available upon request. Tissue samples were provided by the Cooperative Human Tissue Network (CHTN), which is funded by the National Cancer Institute. Other investigators may have received specimens from the same subjects.

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
