# Peer review of "Somatic Hypomethylation of Pericentromeric SST1 Repeats and Tetraploidization in Human Colorectal Cancer Cells"

_cancers, 2021, doi:10.3390/cancers13215353_

Round 1
Reviewer 1 Report
The authors observe, characterise and quantify the phenomenon of LS174T colorectal cancer cells undergoing spontaneous tetraploidization in vitro and the association with demethylation of SST1 repetitive elements. In tumours with severe demethylation of SS1, this is also associated with mutations in TP53. They present a mathematical model for the proportion of diploid and tetraploid cells in a mixed population and a model for cell survival after spontaneous tetraploidization. In the discussion, the authors postulate that the correlation between LS174T SST1 demethylation and spontaneous tetraploidization is causal.
Comments:
Introduction
“Cancer cells exhibit an accelerated epigenetic aging pace [6], leading to a substantial loss of genome-wide methylation (global hypomethylation) and aberrant hypermethylation of some particular loci [7,8].”
The assertion, “…leading to a substantial loss of genome-wide methylation” implies causation. It is unclear whether advanced epigenetic aging is causative or a correlate. I would suggest a correlate.
Horvath-style Epigenetic aging is an elastic net model of CpG correlating with chronological age. These coefficients are both negative and positive. The “epiTOC” (Epigenetic Timer Of Cancer) clock models the *increase* of Polycomb group target (PCGT) promoter CpGs with age.
At the end of Section 1.1, it would be good to comment on the methylation differences and chromosomal stability observed between distal-enriched MSS CIN tumours and the proximal-enriched hypermutator MSI tumours and CIMP tumours. Consideration of changes in global hypomethylation, patient age and defects in chromosomal segregation should minimally be segregated by these groups. CIMP in particular is enriched in older women but does not exhibit the typical methylation patterns observed in the other subgroups.
In the introduction, it would also pay to discuss late replication timing and the association with hypomethylation, laminar-associated domains (LADs) and chromosomal rearrangements. One example is the work of Du 2019 (https://doi.org/10.1038/s41467-019-08302-1).
Results
Section 3.1. It would be useful to further highlight the molecular characteristics of LS174T (CIMP-, MLH1 methylated hypermutator, MSI) Mouradov 2014 (https://doi.org/10.1158/0008-5472.CAN-14-0013)
Sections 3.3, 3.4. “… approximately 2.67% of the mitosis resulted in spontaneous tetraploidization”. Of relevance, the ATCC report that LS-180 (the parent line of LS174T) has a spontaneous tetraploid-like cells occurring at 2.2 percent. https://www.atcc.org/products/cl-187#detailed-product-information
Results Section 3.8. For the TP53 association, it would be useful to also compute Fisher Exact p-values for the 2x2 contingency tables of not demethylated (SST1 delta-RDL < 1) vs Severely demethylated (SST1 delta-RDL > 3). While there is less power due to less n, it removes ambiguity around the measurement error of RDL influencing categorisation of the samples.
Discussion
In the discussion there is a statement, “The causal relationship between DNA de-methylation of SST1 pericentromeric repetitive elements and chromosomal instability is supported by its association with mutational inactivation of p53 or pro-apoptotic proteins in a subset of human primary CRC.”
I am not convinced this reveals a causal relationship. Certainly, it reveals a multi-faceted association. The association of both could well be with a latent causal variable. The abstract is more cautiously worded with regards to causation and the authors also hint earlier in the discussion at a latent causal variable interpretation by suggesting, “… it is possible that the drastic demethylation of SST elements is not per se the direct cause of tetraploidization, but a surrogate diagnostic marker for the underlying epigenetic defect that leads to the genetic alteration.”
Author Response
We thank Reviewer 1 for their work, and for providing constructive comments and suggestions to improve our manuscript.
Introduction
“Cancer cells exhibit an accelerated epigenetic aging pace [6], leading to a substantial loss of genome-wide methylation (global hypomethylation) and aberrant hypermethylation of some particular loci [7,8].”
The assertion, “…leading to a substantial loss of genome-wide methylation” implies causation. It is unclear whether advanced epigenetic aging is causative or a correlate. I would suggest a correlate.
Horvath-style Epigenetic aging is an elastic net model of CpG correlating with chronological age. These coefficients are both negative and positive. The “epiTOC” (Epigenetic Timer Of Cancer) clock models the *increase* of Polycomb group target (PCGT) promoter CpGs with age.
Response: In the new version of the manuscript, we have reduced the mentions of the epigenetic clock in cancer. The original intention was to showcase that some loci become hyper or hypomethylated in all dividing normal tissues, and that these cancer-unrelated methylation changes could be later detected in tumor cells due to clonal expansion together with cancer-specific alterations that, among other causes, could result from an accelerated epigenetic clock. We later indicated that SST1 hypomethylation was NOT one of these age-associated methylation changes but a cancer-associated phenomenon. However, in a more cautious read of our introduction after the comments of both referees, we agree that it could be misleading, suggesting the wrong idea that most DNA methylation changes in cancer would be due to spurious errors associated with an accelerated “aging”.
We have rewritten the introduction to better present the multifactorial nature of the methylation changes found in cancer, included some mechanisms driving coordinated changes in cis and trans, and relevant references that we think will provide excellent reading material for the general reader to better understand the complexity of the topic.
---------
At the end of Section 1.1, it would be good to comment on the methylation differences and chromosomal stability observed between distal-enriched MSS CIN tumours and the proximal-enriched hypermutator MSI tumours and CIMP tumours. Consideration of changes in global hypomethylation, patient age and defects in chromosomal segregation should minimally be segregated by these groups. CIMP in particular is enriched in older women but does not exhibit the typical methylation patterns observed in the other subgroups.
In the introduction, it would also pay to discuss late replication timing and the association with hypomethylation, laminar-associated domains (LADs) and chromosomal rearrangements. One example is the work of Du 2019 (https://doi.org/10.1038/s41467-019-08302-1).
Response: The revised version of the introduction mentions the coordinated methylation found in CRC (CIMP) and the mechanistic link with BRAF mutations (the work of Dr. M.R. Green and colleagues). We also included a description of long range epigenetic silencing (LRES) and activation (LREA), as well as the association of focal hypermethylation with regions of long-range hypomethylation that correlate with late-replication regions of the genome. We also included references to the work Dr. P. Laird, Dr. A. Feinberg/Dr. R. Irizarry, as well as the work of Dr. Susan Clark suggested by the Reviewer.
---------
Results
Section 3.1. It would be useful to further highlight the molecular characteristics of LS174T (CIMP-, MLH1 methylated hypermutator, MSI) Mouradov 2014 (https://doi.org/10.1158/0008-5472.CAN-14-0013)
We included the data and reference in section 3.1
---------
Sections 3.3, 3.4. “… approximately 2.67% of the mitosis resulted in spontaneous tetraploidization”. Of relevance, the ATCC report that LS-180 (the parent line of LS174T) has a spontaneous tetraploid-like cells occurring at 2.2 percent. https://www.atcc.org/products/cl-187#detailed-product-information
Response: We have included this information in section 3.3
---------
Results Section 3.8. For the TP53 association, it would be useful to also compute Fisher Exact p-values for the 2x2 contingency tables of not demethylated (SST1 delta-RDL < 1) vs Severely demethylated (SST1 delta-RDL > 3). While there is less power due to less n, it removes ambiguity around the measurement error of RDL influencing categorisation of the samples.
Response: The p-value of that comparison is 0.0037. We have not included it in table 1, but we mention the comparison and p-value in the revised version of section 3.8 (now 3.7) including part of the Referee’s commentary.
---------
Discussion
In the discussion there is a statement, “The causal relationship between DNA de-methylation of SST1 pericentromeric repetitive elements and chromosomal instability is supported by its association with mutational inactivation of p53 or pro-apoptotic proteins in a subset of human primary CRC.”
I am not convinced this reveals a causal relationship. Certainly, it reveals a multi-faceted association. The association of both could well be with a latent causal variable. The abstract is more cautiously worded with regards to causation and the authors also hint earlier in the discussion at a latent causal variable interpretation by suggesting, “… it is possible that the drastic demethylation of SST elements is not per se the direct cause of tetraploidization, but a surrogate diagnostic marker for the underlying epigenetic defect that leads to the genetic alteration.”
Response: We agree with the Reviewer and apologize for the overreaching statement. While our hypothesis is that there is a causal relationship between SST1 demethylation and CIN that we will try to unravel in future research, our current data does not warrant such a categorical statement. In the revised version of the manuscript, we have tune down the statement.
---------
Reviewer 2 Report
The manuscript by Gonzalez et al. describes the interesting topic of cancer-linked hypomethylation of an acrocentric DNA repeat sequence and its relationship to quantitative chromosomal abnormalities in colorectal cancer cells. However, the main findings are obscured by imprecise language, much wordiness and needless repetition, and overgeneralizations. The manuscript should be shortened (almost halved) and edited throughout. The appropriateness of references and their descriptions should be checked. The Discussion should be entirely rewritten. The authors should consider revising the title of the manuscript because, as they admit in the Discussion, there is no known mechanistic basis for acrosomal epigenetic abnormalities to lead to tetraploidy. They should discuss whether there were gross chromosomal abnormalities in acrocentric chromosomes.
Abstract
Omit “more” and use “strong” or some other term but not the inappropriate “severe” found throughout the manuscript. It is also unclear in the text what the authors definition of “severe” or “strong’ demethylation is.
Introduction
1.1. There is oversimplification of global tissue-specific and age-specific DNA methylation differences, which require more specific and more recent references. For example, there is not ~90% methylation in many different normal human cell types.
There are many places in the manuscript where imprecise descriptions are given that can lead to confusion for the general reader, e.g., “substantial loss of genome wide methylation” (how much and relative to what?).
The authors overgeneralize about DNA methylation losses leading to chromosomal rearrangement. They do not mention that pericentromeric DNA loses much methylation in the ICF syndrome and in many cancers relative to similar normal tissues but this does not lead to proportional changes in ploidy nor to widespread disruption of “mitotic processes,” only to significantly increased frequencies of gross chromosomal abnormalities and only in certain cell types.
The authors refer to cell cycling/aging-related losses in DNA methylation in cancer without noting the critical contribution of alterations in activity of DNMTs and TET enzymes.
They also omit the important contribution of gene region and gene-neighborhood hypomethylation to cancer by direct effects on transcription and effects on chromatin structure.
1.2. The authors omit the previous finding that SST1 sequences can also become hypermethylated in cancer relative to similar normal tissues.
Results
3.1. “Among 14 CRC cell lines, LST174T cells exhibited the highest level of SST1 demethylation and also highest range of variation of SST1 methylation.“
The authors should be specific about the range of variation and level of demethylation. It is critical that they indicate demethylation relative to what standard sample here and elsewhere in the MS when they refer to hypomethylation.
They should be more specific in terminology by stating “near-diploid,” not diploid.
3.2 and 3.3.
They should delete filler or colloquial phrases or descriptions such as: “We intended to subject” “that we reckoned,” “To expedite the process,” and “An obvious interpretation.” They should use “Normal-sized” rather than “regular.”
They should not give more significant figures than warranted by the data, e.g., “2.67% of mitoses.”
“The difference in nuclei size among clones suggested that the parental cell line was a mixed population of small and large nuclei cells…”
The above statement does not “suggest” but rather “indicates,” and this phrase is an example of the wordiness and lack of scientific precision in wording found in the manuscript. Similarly, 3.2 and 3.3 should be combined and shortened.
3.4
“The original publication of the isolation of LS174T reported the presence of 6.1% of tetraploid cells [59]. The higher incidence of polyploidy in our cultures of LS174 cells…”
Shorten to: “The higher incidence of polyploidy in our cultures of LS174 ( %) than in a previous study (Ref & %) is likely due…”
The meaning of the last sentence in this section is unclear.
3.5 Remove “All together, these results show that” and replace with “Therefore,”
3.6 The precent differences in gene expression need to account for multiple testing. Giving p-values is not sufficient.
3.7 Paragraph 1 is very wordy and unnecessarily repeating Methods instead of a brief reference to it.
Paragraph 2.
“tumors overwhelmingly exhibited higher demethylation than normal tis[1]sues (paired t-test p=6x10-23, figure 4D).” Normal tissues cannot exhibit demethylation if they are the standard to which cancers are being compared. Also, which normal tissues are referenced?
Paragraph 3.
“Genome-wide methylation profiling of 30 CRCs and their matching normal samples with Illumina HM450K arrays showed that LINE-1 somatic demethylation correlated with genome-wide demethylation, particularly in CpG sites not associated with CpG islands (those located 2000bp or farther apart from their closest CpG island).”
Where are the data and the significance for this correlation? What does “particularly’ mean? Did you include or exclude CGI sites?
“Therefore, we interpret our data as indicative…”
Instead use “Our findings indicate that cancer-associated SST1 demethylation is occurring in a background of global demethylation.”
3.8
“vs. rest” Use “vs. the other cancers”
Discussion
There is very much redundancy with the Introduction and Results rather than a discussion of the importance of the findings.
Remove imprecise and colloquial statements like “can only thrive”
The discussion tends to overgeneralize the implications of the results as well as being wordy: “In our case there is an added uncertainty on how to explain that demethylation of SST1 elements, that are local[1]ized in only some acrocentric chromosomes may lead to tetraploidization.”
This is followed by a contradictory statement. “The repetitive nature of SST1 elements makes it difficult to determine the precise chromosomal location of those elements that undergo demethylation in cancer.”
The next sentence states that demethylation of different repeats can have different consequences, as if this is a conclusion rather than an obvious limitation.
The following sentence refers to a lncRNA without stating its genomic origin: “We recently reported that demethyla-tion of SST1 precedes the expression of a long non-coding RNA that forms a perinucleolar aggregate with RNA binding proteins.”
Author Response
We thank reviewer #2 for their work, and for providing constructive comments and suggestions to improve our manuscript.
The manuscript by Gonzalez et al. describes the interesting topic of cancer-linked hypomethylation of an acrocentric DNA repeat sequence and its relationship to quantitative chromosomal abnormalities in colorectal cancer cells. However, the main findings are obscured by imprecise language, much wordiness and needless repetition, and overgeneralizations. The manuscript should be shortened (almost halved) and edited throughout. The appropriateness of references and their descriptions should be checked. The Discussion should be entirely rewritten.
Response: We have rewritten large parts of the manuscript, trying to be more precise. This has resulted in a slightly longer introduction, but significantly shorter results and discussion sections (see details below).
---------
The authors should consider revising the title of the manuscript because, as they admit in the Discussion, there is no known mechanistic basis for acrosomal epigenetic abnormalities to lead to tetraploidy.
Response: In our view, the title does not imply a mechanistic association, just indicates an association between these two phenomena.
----------
They should discuss whether there were gross chromosomal abnormalities in acrocentric chromosomes.
Repsonse: The karyotyping did not reveal gross chromosomal abnormalities in acrocentric chromosomes in any of the near-tetraploid LS174T subclones. We have explicitly indicated this observation in the discussion.
----------
Abstract
Omit “more” and use “strong” or some other term but not the inappropriate “severe” found throughout the manuscript.
Response: In previous publications (PMIDs 31867127, 29912433) and presentations to congresses (AACR 102nd Annual Meeting 2011, 10.1158/1538-7445.AM2011-2792; AACR Annual Meeting 2014, 10.1158/1538-7445.AM2014-1381), we employed the term “severe” with its meaning of “of a great degree” when referring to primary tumors exhibiting strong demethylation of SST1 elements (compared with matching normal tissues).
Nevertheless, in the rewritten version of the paper, we have avoided the term “severe”, employing the term “strong” instead.
----------
It is also unclear in the text what the authors definition of “severe” or “strong’ demethylation is.
Response: In the original analysis (PMID 31867127) we employed bisulfite sequencing to calculate the percentage of methylation in primary tumors and their matching normal samples. The difference in average methylation between tumor and matching normal sample was named somatic demethylation. Based on the distribution of somatic demethylation values, tumors with average SST1 methylation level between 5% to 10% lower than their matching normal samples were considered moderately demethylated, and those with a difference larger than 10% were considered “severely” demethylated. We have indicated these thresholds in the revised introduction. In this work, we used a much faster and economical QPCR-based technology to measure SST1 methylation. This technology does not provide a percentage of methylation, but the proportion of unmethylated SST1 vs methylation-independent amplifiable elements in a sample, normalized vs that proportion in DLD1 (RDL, expressed after applying log2 transformation). The RDL difference between tumors and matching normal samples estimates somatic hypo or hypermethylation (somatic demethylation value, or ΔRDL). As we indicated in the manuscript, SST1 RDL in tumors correlated with that in matching normal tissues. There were nine tumors that deviated from this correlation (seven above and two below the 95% CI, figures 4A and 4B). The seven above the 95% CI were considered strongly hypomethylated and all of them exhibited SST1 ΔRDL >3, indicating that the proportion of unmethylated SST1 elements was >8 times (23) higher in the tumor than in their matching normal tissue. Albeit this classification can be somehow arbitrary, note that we have also performed statistical analyses using ΔRDL as a continuous variable without applying a categorical classification, obtaining consistent results regarding the association of SST1 ΔRDL with TP53 mutations.
----------
Introduction
1.1. There is oversimplification of global tissue-specific and age-specific DNA methylation differences, which require more specific and more recent references. For example, there is not ~90% methylation in many different normal human cell types.
There are many places in the manuscript where imprecise descriptions are given that can lead to confusion for the general reader, e.g., “substantial loss of genome wide methylation” (how much and relative to what?).
The authors overgeneralize about DNA methylation losses leading to chromosomal rearrangement. They do not mention that pericentromeric DNA loses much methylation in the ICF syndrome and in many cancers relative to similar normal tissues but this does not lead to proportional changes in ploidy nor to widespread disruption of “mitotic processes,” only to significantly increased frequencies of gross chromosomal abnormalities and only in certain cell types.
Response: we have included all these suggestions in the revised version of the manuscript. As the Referee correctly indicates, SST1 sequences are hypomethylated in lymphocytes from ICF syndrome patients, associated with increased chromosomal abnormalities in chromosomes 1, 16, and 9. We have not found alterations in these chromosomes in LS174T cells, though. A plausible explanation for the difference between ICF cells and LS174T is that the chromosomal alterations found in ICF cells might be caused by the widespread hypomethylation of other centromere-related sequences (Sat2 and Sat3) including those located in chromosomes 1, 16, and 9. Also, our data indicate that in CRC SST1 hypomethylation associates with the inactivation of p38-p53-p21 pathway that, to the best of our knowledge, is functional in ICF cells. This might explain why tetraploidization is not found in ICF cells. We discuss these ideas in the revised version of the manuscript.
-----------
The authors refer to cell cycling/aging-related losses in DNA methylation in cancer without noting the critical contribution of alterations in activity of DNMTs and TET enzymes.
Response: In the new version, we mention that “DNA methylation is a dynamic process mediated by the activity of DNA methyltransferases and TET enzymes that are involved in deposition and removal of DNA methylation, respectively.”
We apologize for the imprecise wording of our previous version. The original intention was to showcase that some loci become hyper or hypomethylated in all dividing normal tissues, and that these cancer-unrelated methylation changes can be later detected in tumor cells due to clonal expansion together with cancer-specific alterations that, among other causes, could result from an accelerated epigenetic clock. We later indicated that SST1 hypomethylation was NOT one of these age-associated methylation changes but a cancer-associated phenomenon. However, in a more cautious read of our introduction after the comments of both referees, we agree that it could be misleading, suggesting the wrong idea that most DNA methylation changes in cancer would be due to spurious errors associated with an accelerated “aging”.
We have rewritten the introduction to better explain the multifactorial nature and complexity of cancer-related DNA methylation alterations.
-----------
They also omit the important contribution of gene region and gene-neighborhood hypomethylation to cancer by direct effects on transcription and effects on chromatin structure.
Response: In the revised version of the introduction, we have included a new paragraph mentioning gene region and gene-neighborhood effects of DNA methylation, and several mechanisms involved in cis and trans coordinated methylation alterations (LRES, LREA, focal hypermethylation within long-range hypomethylation regions associated with late-replication domains, CIMP), providing relevant references to facilitate the general reader to gain a deeper understanding of the topic.
--------
1.2. The authors omit the previous finding that SST1 sequences can also become hypermethylated in cancer relative to similar normal tissues.
Response: In the revised version we explicitly indicate that these sequences have been found hypermethylated in some types of cancers, as well as providing the proportion of CRCs that we found with moderate or strong hypomethylaton and with moderate hypermethylation: “Around 22% of the colorectal cancers (CRC) were hypomethylated (>5% lower methylation compared with matched normal tissue), including 7% of CRCs that exhibited strong hypomethylation of SST1 elements (≥10% demethylation compared with matched normal tissue), that we named “severe” hypomethylation. The majority of the CRCs (78%) exhibited no hypomethylation, with a few of them (~5%) exhibiting moderate hypermethylation (>5% hypermethylation compared to matched normal tissue)”.
----------
Results
3.1. “Among 14 CRC cell lines, LST174T cells exhibited the highest level of SST1 demethylation and also highest range of variation of SST1 methylation.“
The authors should be specific about the range of variation and level of demethylation. It is critical that they indicate demethylation relative to what standard sample here and elsewhere in the MS when they refer to hypomethylation.
Response: we reworded this sentence to “Among 14 CRC cell lines, LST174T cells exhibited the lowest average level (55±25%) and the highest range ([19%-88%]) of SST1 methylation“. We generally consider demethylation as the loss of methylation in the tumor relative to the matching non-tumoral colonic tissues (as shown in figures 4A and 4B). Albeit the normal tissue of the patient from which LS174T cells were obtained is not available, none of the 80 non-tumoral colonic samples that we analyzed in our previous publication, nor any of the 14 CRC cell lines, exhibited such a low level of methylation. According to our new MS-QPCR data the SST1 RDL of LS174T was > 6, indicating that the proportion of unmethylated SST1 sequences is over 64 times (2^6) higher than that in DLD1 (supplementary figure S5). This RDL value is only surpassed by 3 of the strongly demethylated CRCs and none of the 148 normal tissues analyzed in this manuscript. Thus, we think it is reasonable to infer that LS174T cells had in fact undergone somatic demethylation. Nevertheless, we avoided the term “demethylated” when referring to LS174T, and used “low methylation” instead.
---------
They should be more specific in terminology by stating “near-diploid,” not diploid.
In the revised version we employ near-diploid or near-tetraploid when referring to LS174T cells.
----------
3.2 and 3.3.
They should delete filler or colloquial phrases or descriptions such as: “We intended to subject” “that we reckoned,” “To expedite the process,” and “An obvious interpretation.” They should use “Normal-sized” rather than “regular.”
Response: Corrected as suggested.
---------
They should not give more significant figures than warranted by the data, e.g., “2.67% of mitoses.”
Repsonse: Corrected to “around 3%”
----------
“The difference in nuclei size among clones suggested that the parental cell line was a mixed population of small and large nuclei cells…”
The above statement does not “suggest” but rather “indicates,” and this phrase is an example of the wordiness and lack of scientific precision in wording found in the manuscript.
Similarly, 3.2 and 3.3 should be combined and shortened.
Response: In the revised version we have shortened and combined sections 3.2 & 3.3, and removed the abovementioned statement.
---------
3.4
“The original publication of the isolation of LS174T reported the presence of 6.1% of tetraploid cells [59]. The higher incidence of polyploidy in our cultures of LS174 cells…”
Shorten to: “The higher incidence of polyploidy in our cultures of LS174 ( %) than in a previous study (Ref & %) is likely due…”
Repsonse: Corrected as suggested.
---------
The meaning of the last sentence in this section is unclear.
Response: We rewrote the sentence as “All fifteen subclones exhibited very high chromosomal instability and ploidy variability, and no distinctive groups based on their nuclei area or ploidy could be observed in the time frame of the cultures (data not shown).”
-----------
3.5 Remove “All together, these results show that” and replace with “Therefore,”
Response: Corrected as suggested.
------------
3.6 The precent differences in gene expression need to account for multiple testing. Giving p-values is not sufficient.
Response: The experiment included 4 near-tetraploid and 4 near-diploid LS174T subclones that exhibited extremely similar transcriptome profiles (all pair-wise correlations were between 0.96 and 0.99). Due to the overall high similarity between clones regardless of their ploidy, very few genes exhibited statistically significant differences (84/21448 ≈ 0.4% with p < 0.01) even before FDR multiple testing correction. None of them exhibited a statistically significant difference after FDR correction (supplementary figure S7). In the revised main text, we removed all mentions to HDLBP, TBC1D16, and GNAO1, just indicating “Near-diploid and near-tetraploid LS174T subclones exhibited similar transcriptional profiles (supplemental figure S7)”. In figure S7 we indicate “In red, genes that reached statistical significance (0.39%, n=84) before FDR multi-hypothesis testing correction. No gene exhibited a statistically significant difference after FDR correction”. The GSEA analysis (figure S8), on the other hand, included FDR and FWER multiple testing corrections.
---------
3.7 Paragraph 1 is very wordy and unnecessarily repeating Methods instead of a brief reference to it.
Response: We have shortened paragraph 1.
----------
Paragraph 2.
“tumors overwhelmingly exhibited higher demethylation than normal tis[1]sues (paired t-test p=6x10-23, figure 4D).” Normal tissues cannot exhibit demethylation if they are the standard to which cancers are being compared. Also, which normal tissues are referenced?
Response: rephrased as “most tumors exhibited lower levels of methylation (higher RDL) than their matching normal mucosa samples”. The matching normal mucosa samples are the adjacent non-tumoral colonic biopsies taken at the time of operation.
----------
Paragraph 3.
“Genome-wide methylation profiling of 30 CRCs and their matching normal samples with Illumina HM450K arrays showed that LINE-1 somatic demethylation correlated with genome-wide demethylation, particularly in CpG sites not associated with CpG islands (those located 2000bp or farther apart from their closest CpG island).”
Where are the data and the significance for this correlation? What does “particularly’ mean? Did you include or exclude CGI sites?
Response: The analysis was presented in supplemental figure 10. In the original analysis, we investigated the correlation between LINE1 and SST1 RDL with hyper- and hypomethylation measured using the HM450K arrays. The probes of these arrays were classified into 4 categories depending to their distance to the nearest CpG island: open sea (>4000bp, n=154,993), shelf (2000-4000bp, n=41,770), shore (<2000bp, n=103,803) or island (inside CpG island, n=142,002). We analyzed the correlations of hyper- and hypomethylation in these regions with the SST1 and LINE1 RDL.
Rephrased as “Genome-wide methylation profiling of 30 CRCs and their matching normal samples with Illumina HM450K arrays showed that LINE-1 somatic demethylation correlated with genome-wide demethylation (r=0.68, p=4.4x10-5, supplemental figure S10C). When considering only CpG sites not associated with CpG islands (those located > 2000bp apart from their closest CpG island) the correlation remained high r=0.69, p=3.6x10-5 (supplemental figure S10C). This association was expected since LINE-1 methylation is widely considered a marker of global genome methylation. SST1 demethylation, on the other hand, did not correlate with genome-wide demethylation regardless of the analyzed subset of CpG sites (supplemental figure S10C). ”
-------
“Therefore, we interpret our data as indicative…”
Instead use “Our findings indicate that cancer-associated SST1 demethylation is occurring in a background of global demethylation.”
Response: Corrected as suggested.
--------
3.8
“vs. rest” Use “vs. the other cancers”
Response: Corrected as suggested.
----------
Discussion
There is very much redundancy with the Introduction and Results rather than a discussion of the importance of the findings.
Response: we have shortened the Discussion to reduce the overlapping with Introduction and Results.
--------
Remove imprecise and colloquial statements like “can only thrive”
Response: We have rephrased that sentence as “Therefore, cells that undergo tetraploidization would require the impairment of the p38-p53-p21 pathway either by mutations in TP53 or by inactivation of essential downstream effectors such as BAX to evade cell cycle arrest”.
--------
The discussion tends to overgeneralize the implications of the results as well as being wordy: “In our case there is an added uncertainty on how to explain that demethylation of SST1 elements, that are local[1]ized in only some acrocentric chromosomes may lead to tetraploidization.”
This is followed by a contradictory statement. “The repetitive nature of SST1 elements makes it difficult to determine the precise chromosomal location of those elements that undergo demethylation in cancer.”
The next sentence states that demethylation of different repeats can have different consequences, as if this is a conclusion rather than an obvious limitation.
Response: We did not intend to present that as a conclusion but, as the referee indicates, a limitation of our study. That is why we mentioned difficulty to identify which SST1 elements are undergoing hypomethylation. We do not know whether the demethylation targets just a few SST1 elements or most of them, nor the precise location of the SST1 elements that undergo demethylation.
We have rephrased those senteces as “A limitation of our study is that, due to their repetitive nature, we could not determine the precise chromosomal location of the SST1 elements that undergo somatic demethylation. We cannot rule out that demethylation of some of these elements, but no others, might have different phenotypic effects.”
-----------
The following sentence refers to a lncRNA without stating its genomic origin: “We recently reported that demethyla-tion of SST1 precedes the expression of a long non-coding RNA that forms a perinucleolar aggregate with RNA binding proteins.”
In the revised version we mention the SST1-derived lncRNA in a different paragraph, as follows: “We recently reported that demethylation of SST1 was associated with the expression of TNBL, a long non-coding RNA originating from these sequences stable and throughout the mitotic cycle, that formed a perinucleolar aggregate with RNA binding proteins. The function of this lncRNA or its putative effect on genome duplication fidelity is still unknown.”